# Large Splenic Abscess Caused by Non-Typhoidal *Salmonella* in a Healthy Child Treated with Percutaneous Drainage

**DOI:** 10.3390/children7080088

**Published:** 2020-08-03

**Authors:** Hyun Woo Lee, Seung Beom Han

**Affiliations:** Department of Pediatrics, College of Medicine, The Catholic University of Korea, Seoul 06591, Korea; zzangwoo91@naver.com

**Keywords:** child, *Salmonella*, spleen abscess

## Abstract

Splenic abscess occurs very rarely in healthy children. Although typhoid fever was the leading cause of splenic abscess in the pre-antibiotic era, *Salmonella* spp. remain to be the major pathogens causing splenic abscess, with an increasing worldwide frequency of splenic abscess due to non-typhoidal *Salmonella* infection. Here, we report the case of a 12-year-old boy, who was presumably diagnosed with acute gastroenteritis on admission and eventually diagnosed with a large splenic abscess (maximum diameter, 14.5 cm) caused by non-typhoidal *Salmonella*. Although splenectomy has been considered in cases of large splenic abscesses, the patient was treated with antibiotics and ultrasonography-guided percutaneous drainage. A detailed physical examination and appropriate imaging studies are necessary for the early diagnosis of extra-intestinal complications of non-typhoidal *Salmonella* enteritis. For treatment, percutaneous drainage, rather than splenectomy, can be used in large splenic abscesses.

## 1. Introduction

Isolated splenic abscess occurs very rarely in children. In the pre-antibiotic era, typhoid fever caused by *Salmonella enterica* serovar Typhi frequently predisposed individuals to splenic abscess, although its occurrence decreased with the introduction of antibiotic therapy [1]. However, in the endemic area of typhoid fever, *S.* ser. Typhi still remains a leading cause of splenic abscess [2]. Currently, the leading cause of splenic abscess is infective endocarditis, with *Streptococcus* and *Staphylococcus* being the most common causative agents [3]. However, *Salmonella* spp. are still considered major pathogens of splenic abscess, as evidenced by the increasing worldwide frequency of splenic abscess due to non-typhoidal *Salmonella* (NTS) infection rather than *S*. ser. Typhi infection [4].

In the current report, we diagnosed an isolated large splenic abscess caused by NTS infection in a boy who was presumably diagnosed with acute gastroenteritis (AGE) on admission. Although the patient had a large splenic abscess (maximum diameter, 14.5 cm), he was treated with antibiotics and ultrasonography (US)-guided percutaneous drainage rather than splenectomy.

## 2. Case Presentation

The case was a 12-year-old boy who presented with fever, abdominal pain lasting for three days, and diarrhea that developed one day prior. Vomiting occurred during the first day of fever. He had frequently experienced recurrent abdominal pain and vomiting lasting for 1 or 2 days; however, he reported no recent gastrointestinal (GI) symptoms for the previous month. Moreover, he had not recently traveled abroad. On admission, his vital signs were as follows: blood pressure, 120/60 mmHg; body temperature, 39.7 °C; heart rate, 72 beats/min; respiratory rate, 24 breaths/min. He looked acutely ill, and a soft and flat abdomen with tenderness in the epigastric and left upper quadrant areas was noted on physical examination. Laboratory tests revealed white blood cell (WBC) count of 13,000/μL (neutrophils 85%, lymphocytes 6%), hemoglobin of 11.8 g/dL, platelet count of 177,000/μL, erythrocyte sedimentation rate of 75 mm/h, and C-reactive protein (CRP) level of 31.00 mg/dL. The chest X-ray showed no abnormality; however, an abdominal X-ray showed a diffusely increased soft-tissue density suggestive of splenomegaly in the left upper quadrant. Based on his symptoms, physical examination, and laboratory results, empirical antibiotic therapy for a presumptive diagnosis of bacterial AGE was initiated (cefotaxime 150 mg/kg/day). On the admission day, abdominal computed tomography (CT) was performed to evaluate suspicious splenomegaly found on the abdominal X-ray and possible intra-abdominal complications of AGE. Results of the CT scan showed a large abscess (maximum diameter, 14.5 cm) in the upper pole of the spleen (Figure 1A). 

Intravenous metronidazole (40 mg/kg/day) was added to the cefotaxime therapy for possible amoebic abscess. On the second hospital day (HD #2), US-guided percutaneous drainage of the splenic abscess was performed due to an impending rupture. Anchovy paste-colored pus was drained (840 mL), and the laboratory examination of pus revealed a WBC count of 20,800/μL (neutrophils 95%, lymphocytes 2%), red blood cell count of 20,750/μL, and protein and glucose levels of 5.5 g/dL and 0 mg/dL, respectively. With antibiotic therapy, diarrhea and abdominal pain resolved on HD #3 and #5, respectively. The blood and urine samples that were collected upon admission were sterile. In contrast, *Salmonella* spp. group D, which was susceptible to ampicillin and cefotaxime, was detected in the stool and drained splenic abscess fluid samples. The fever disappeared on HD #7. On HD #11, when the amount of drained pus fluid decreased to less than 100 mL/day, an abdominal CT was again performed, indicating that the splenic abscess had decreased in size (Figure 1B). Metronidazole was administered for 2 weeks due to a possible combined infection of intra-abdominal anaerobes or amoebic abscess. The stool and splenic abscess fluid were analyzed at the Korea Centers for Disease Control and Prevention, reporting negative results of polymerase chain reaction tests for *Entamoeba histolytica* on HD #19. A maculopapular skin rash consistent with a drug rash developed on HD #21; therefore, the administered cefotaxime was changed to oral amoxicillin/clavulanate. The amount of drained abscess fluid decreased to less than 30 mL/day on HD #22; abdominal US on this day showed no pus in the spleen, thus the drainage catheter was removed. The patient was discharged from the hospital on HD #26 with continuing oral amoxicillin/clavulanate therapy. Echocardiography performed during hospitalization showed no evidence of infective endocarditis, and lymphocyte subset counts as well as immunoglobulin and complement levels were within the normal ranges. Abdominal US repeated in the outpatient department (OPD) 8 days after discharge showed a splenic pseudocyst with a maximum diameter of 4.1 cm. Antibiotic therapy was discontinued in the OPD 6 weeks after the diagnosis of splenic abscess. Repeated abdominal US showed a persistent splenic pseudocyst without a change in size at 7 months after the end of the antibiotic therapy.

## 3. Discussion

The genus *Salmonella* is divided into two species, *S. enterica* and *S. bongori*, and *S. enterica* is further subdivided into typhoidal *Salmonella* and NTS [4]. Typhoidal *Salmonella*, including *S*. ser. Tyhpi and *S*. ser. Paratyphi, causes systemic infection (typhoid fever), while NTS causes self-limiting AGE in most cases [4]. However, NTS may cause invasive and extra-intestinal infections, especially in immunocompromised patients [4]. *S*. ser. Typhi was the most common pathogen of splenic abscess in the pre-antibiotic era; however, NTS became a more frequent cause of splenic abscess than *S*. ser. Typhi after the introduction of antibiotic therapy [1]. Nevertheless, invasive and extra-intestinal infections by NTS occur very rarely, with the ratio of invasive to enteric infections by NTS reported to be 1:28 [5]. In particular, the invasive NTS infection rate is lower in Asia and Oceania than in other areas of the world, with the ratio of invasive to enteric NTS infection reported to be 1:3851 [5]. Moreover, splenic abscess composes a very small proportion of invasive NTS infections. Ispahani et al. reported only one patient with splenic abscess among 25 patients with extra-intestinal NTS infection [6]. Therefore, predicting the presence of splenic abscess in a patient with NTS enteritis is very challenging. In fact, we did not recognize splenomegaly on the initial physical examination of our patient because a large abscess located at the upper pole of the spleen was softly palpated in the left subcostal area. However, the abdominal X-ray showed an area of increased density suggestive of splenomegaly, while a high blood CRP level implied complicated AGE. Both results led to the performance of an abdominal CT, confirming a large splenic abscess. On repeated physical examination after abdominal CT, the lower margin of the spleen was palpated at 3 cm below the left costal margin. A detailed physical examination and appropriate imaging studies are mandatory to identify such rare complications of NTS enteric infections.

Although our patient complained of recurrent abdominal pain and vomiting, his previous GI symptoms improved within 1–2 days, but he denied having any recent GI symptoms accompanying the fever within 1 month prior to admission. Previous reports showed that children and adolescents with large splenic abscesses sized 14–23 cm complained of fever and GI symptoms lasting for only 2–3 days [7,8]. Therefore, the large splenic abscess in our patient could have developed rapidly as an acute complication of NTS enteritis. Conversely, considering the negative blood and positive stool culture results in our patient, the previous mild NTS enteritis might have caused transient bacteremia promoting focal splenic abscess and colonic NTS colonization, and the growing untreated splenic abscess might have caused acute symptoms.

Although splenectomy was traditionally recommended for splenic abscess [3], successes of antibiotic therapy with percutaneous aspiration or drainage have been reported [9,10]. Especially in children, splenectomy should be avoided whenever possible due to presumptive immune deficiency following splenectomy, increasing risk of encapsulated bacterial infection, and antibiotic prophylaxis and vaccination for these bacteria. Percutaneous aspiration and drainage were performed for splenic abscesses with ≤10 cm maximum diameter [9,10], while percutaneous drainage was successful in our patient with a large splenic abscess. Therefore, initial percutaneous drainage and antibiotic therapy, rather than splenectomy, can be used to treat large unruptured splenic abscesses. Considering the treatment failures of percutaneous aspiration or short-term drainage in abscesses larger than 10 cm in diameter [8], percutaneous drainage should be maintained as long as needed. Hemorrhage is the most common major complication of splenic intervention, and a massive hemorrhage may require splenectomy [11]. Other major complications, including pneumothorax, pleural empyema, colonic injury, renal injury, fistula formation, and minor complications, including local pain and subcapsular hematoma, can also occur with percutaneous drainage [11]. Although a previous study did not report any complications associated with percutaneous aspiration and drainage in 36 patients with splenic abscesses [9], the patient should be closely monitored for complications, especially during prolonged drainage.

In conclusion, a 12-year-old boy presenting with symptoms of AGE was eventually diagnosed with an isolated large splenic abscess due to NTS. A detailed physical examination and appropriate imaging studies are necessary for the early diagnosis of extra-intestinal complications of NTS enteritis. For treatment, percutaneous drainage, rather than splenectomy, can be used in large splenic abscesses.

## Figures and Tables

**Figure 1 children-07-00088-f001:**
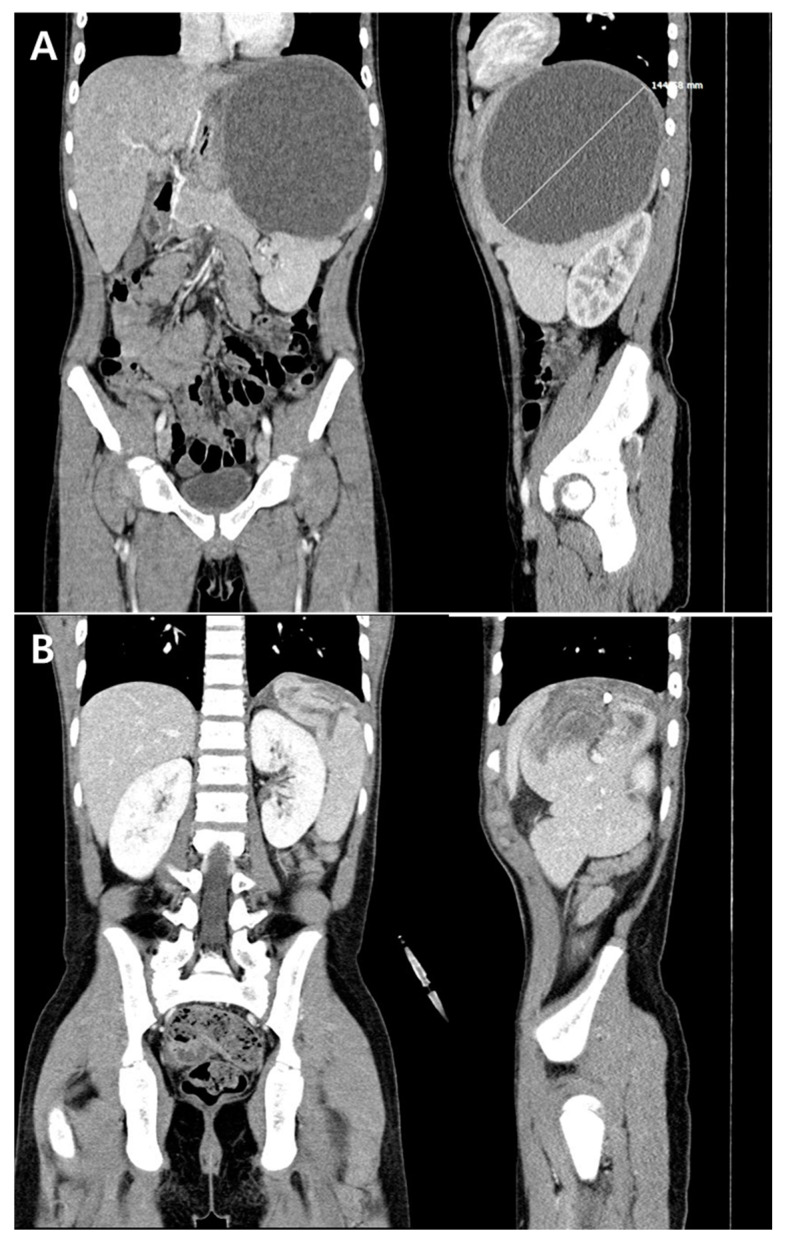
Abdominal computed tomography shows a large abscess with a maximum diameter of 14.5 cm on admission (**A**), and the abscess decreased in size on hospital day (HD) #11 (**B**).

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
