# Peer review of "Large Splenic Abscess Caused by Non-Typhoidal Salmonella in a Healthy Child Treated with Percutaneous Drainage"

_children, 2020, doi:10.3390/children7080088_

Round 1

Reviewer 1 Report

This is an interesting case report of a boy with a large splenic abscess caused by non-typhoidal Salmonella, which was successful treated with antibiotics and ultrasonography guided percutaneous drainage rather than splenectomy.

However, there are parts of the report not completely clear:

1) In the abstract, what does antibiotic splenectomy mean ?

2) The case presentation needs English language revision, i.e. verb tenses, of the history of the present illness and the past medical history.

3) Possible complications of ultrasonography guided percutaneous drainage must be also discussed.

Author Response

Thank you for valuable advice on the manuscript.

Please see the attachment for a point-by-point response.

Reviewer 2 Report

Dear authors, thank you for your interesting work! The text was a little bit unclear in some parts, due to large sentences. Only minor comments from my side, which you can find below:

Introduction: Isolated splenic abscess occurs very rarely in children. In the pre-antibiotic era, typhoid fever caused by Salmonella enterica serova Typhi, frequently predisposed individuals to splenic abscess; although, its occurrence decreased with antibiotic therapy [1]. However, in the endemic area of typhoid fever, Salmonella enterica serova Typhi still remains a leading cause of splenic abcess [3]. Please change the references accordingly Currently, the leading cause of splenic abscess is infective endocarditis with Streptococcus and Staphylococcus being the most common causative agents [2]. However, Salmonella spp. are still considered major pathogens of splenic abscess, as evidenced by the increasing worldwide frequency of splenic abscess due to non-typhoidal Salmonella (NTS) infection rather than S. ser. Typhi infection [4]. In the current report, we diagnosed isolated large splenic abscess caused by NTS infection in a boy that was presumably diagnosed with acute gastroenteritis (AGE) on admission. Although the patient had a large splenic abscess (maximum diameter, 14.5 cm), he was treated with antibiotics and ultrasonography (US)-guided percutaneous drainage rather than splenectomy. 

Line 36: A 12-year-old boy presented with fever, and abdominal pain lasting three days, and diarrhea, that developed one day prior.

Line 49: On the admission day, abdominal computed tomography (CT) was performed to evaluate suspicious splenomegaly found on the abdominal X-ray and possible intra-abdominal complications of AGE. Results from the CT scan showed a large abscess (maximum diameter, 14.5 cm) in the upper pole of the spleen (Figure 1A).

Line 62: The blood and urine samples that were collected upon admission were sterile. In contrast, Salmonella spp. group D, which was susceptible to ampicillin and cefotaxime, was detected in the stool and the drained splenic abscess fluid samples.

Line 64: The fever disappeared on HD #7. On HD #11, when the amount of drained pus fluid decreased to less than 100 mL/day, an abdominal CT was performed, indicating that the splenic abscess had decreased in size (Figure 1B).

Line 66: Metronidazole was administered for 2 weeks due to a possible combined infection of intra-abdominal anaerobes or amoebic abscess. Stool and splenic abscess fluid were analyzed at the Korea Centers for Disease Control and Prevention, reporting negative results of polymerase chain reaction tests for Entamoeba histolytica on HD #19.

Line 73: The patient was discharged from the hospital on HD #26 with continuing oral amoxicillin/clavulanate therapy.

Line 97: Did you mean the following?: However, the abdominal X-ray showed an area of increased density suggestive of splenomegaly while the high blood CRP level implied complicated AGE. Both results led to the performance of an abdomen CT, confirming a large splenic abscess.

Author Response

(The authors gave the same response as above.)
